# Social Network Analysis of COVID-19 Sentiments: 10 Metropolitan Cities in Italy

**DOI:** 10.3390/ijerph19137720

**Published:** 2022-06-23

**Authors:** Gabriela Fernandez, Carol Maione, Harrison Yang, Karenina Zaballa, Norbert Bonnici, Jarai Carter, Brian H. Spitzberg, Chanwoo Jin, Ming-Hsiang Tsou

**Affiliations:** 1Metabolism of Cities Living Lab, Center for Human Dynamics in the Mobile Age, Department of Geography, San Diego State University, San Diego, CA 92182, USA; carol.maione@polimi.it (C.M.); hyang5959@sdsu.edu (H.Y.); nikazaballa@gmail.com (K.Z.); norbert@bonnici.mt (N.B.); jarai.carter@gmail.com (J.C.); spitz@sdsu.edu (B.H.S.); cjin@sdsu.edu (C.J.); mtsou@sdsu.edu (M.-H.T.); 2Department of Management, Economics, and Industrial Engineering, Politecnico di Milano, 20156 Milan, Italy; 3Malta Critical Infrastructure Protection Directorate, 1532 Valletta, Malta; 4Smart Lab, Procter & Gamble, Champaign, IL 61820, USA; 5Department of Communication, San Diego State University, San Diego, CA 92182, USA

**Keywords:** COVID-19, Italy, sentiment analysis, Twitter, fear, anger, joy, metropolitan cities, social network analysis

## Abstract

The pandemic spread rapidly across Italy, putting the region’s health system on the brink of collapse, and generating concern regarding the government’s capacity to respond to the needs of patients considering isolation measures. This study developed a sentiment analysis using millions of Twitter data during the first wave of the COVID-19 pandemic in 10 metropolitan cities in Italy’s (1) north: Milan, Venice, Turin, Bologna; (2) central: Florence, Rome; (3) south: Naples, Bari; and (4) islands: Palermo, Cagliari. Questions addressed are as follows: (1) How did tweet-related sentiments change over the course of the COVID-19 pandemic, and (2) How did sentiments change when lagged with policy shifts and/or specific events? Findings show an assortment of differences and connections across Twitter sentiments (fear, anger, and joy) based on policy measures and geographies during the COVID-19 pandemic. Results can be used by policy makers to quantify the satisfactory level of positive/negative acceptance of decision makers and identify important topics related to COVID-19 policy measures, which can be useful for imposing geographically varying lockdowns and protective measures using historical data.

## 1. Introduction

In 2020, lives changed dramatically due to the outbreak of coronavirus disease, known as SARS-CoV-2 or COVID-19. Since then, the scientific community has shown remarkable success in determining the short- and long-term health impacts of the pandemic, especially those associated with a higher risk of fatality and pre-existing conditions [1,2,3]. Lesser known are the effects on mental health and behaviors linked to lockdown measures and containment policies, fear of contagion and ultimately death, chronic anxiety, and distress, that superimposed unprecedented restrictions on individual freedom. Studies have associated these conditions with an increase in reported psychological disorders and depression, especially across younger groups [4]. Furthermore, when combined with pre-existing conditions, psychological disorders could lead to severe psychiatric consequences and suicidal behavior [5].

Previous studies demonstrate that social media are a gold mine for investigating the societal response to the pandemic outbreak [6,7,8]. During times of emergency, in fact, social media platforms become primary communication channels that ordinary people use to form and share opinions and beliefs around a certain topic and monitor real-time emotions [9,10]. People rely more and more on social media sites such as Twitter or Facebook to connect to the outer world, showcasing emotions and sentiments linked to COVID-19 [11].

This study deals with an extensive analysis of emotions contained in Twitter posts during the first wave of the pandemic. For this study, 4,227,882 million coronavirus-related tweets between 2 March 2020 and 15 June 2020 were collected for 10 Italian cities during the period of March 2020 to June 2020. The research aimed to explore regional variations in the social response to COVID-19 across Italy through the analysis of three sentiment emotions: fear, anger, and joy. The aim of this study was to analyze the national spread of COVID-19 across Italy (north, center, south, and islands). The striking spatial unevenness of COVID-19 suggests that the infection has hit economic core locations harder, and this raises questions about whether, and how, the geography of the disease is connected to the economic base of localities and policy. This study viewed 10 metropolitan cities (i.e., widely affected regions) while analyzing sentiments and perceptions across the north, center, south, and island regions of Italy to identify geospatial patterns during the COVID-19 pandemic, which follow the lines of the local economic landscape, culture, resources, funding, infrastructure, policy, and spatial economic forces. The COVID-19 restrictions and containment measures aimed at preventing the mobility and interaction of people inside and across these areas and included, for instance, the suspension of social gatherings, cultural and religious activities, and limitations on restaurants, just to name a few, and had an impact on the people’s well-being, perceptions, and sentiments across the country. This study aimed at presenting a data-driven approach to exploring the spatial-temporal patterns of the pandemic across Italy.

### Related Work

Digital communication media are evolving much faster than our theoretical frameworks for conceptualizing their processes and effects, although the need for theory in social media and big data remains important. From user-generated content on social media, studies have analyzed the public’s thoughts and sentiments on health status, concerns, panic, and awareness related to COVID-19, which can ultimately assist in developing health intervention strategies and designing effective campaigns based on public perceptions [12]. Scholars have taken similar approaches to develop sentiment analysis using Twitter data around the world. One example study of a sentiment analysis and topic modeling research focused on the public perception of the COVID-19 pandemic on Twitter. The study aimed to increase understanding of public awareness of the pandemic trends and uncover meaningful themes of concern posted by Twitter users. The analyses included frequency of keywords, sentiment analysis, and topic modeling to identify and explore discussion topics over time [13]. This work only shows the negative sentiment from some particular topics without analyzing any model in time intervals and is devoid of any sentiment model and analysis. Additionally, the appearance of opinions showed that the approach is stable and viable for understanding public opinion. Other scholars have examined sentiments evoked in tweets only with the judgment of some emerging keywords about COVID-19, examining the top trending topics over time [14]. Furthermore, scholars have presented an issue surrounding public sentiment, leading to the testimony of growth in fear sentiment and negative sentiment [15]. 

Overall, this study focused on Twitter sentiment analysis (fear, anger, and joy) based on geography across Italy during the COVID-19 pandemic and the impact of policy measures/cases, and deaths during the pandemic. The details of the proposed research workflow model are discussed in the next section.

## 2. Materials and Methods

### 2.1. Study Area

Italy’s first confirmed case of COVID-19 was reported on 20 February 2020 in the Province of Lodi, Lombardy. The number of COVID-19 cases escalated quickly, with hundreds of positive cases registered within a few days (with an average mortality rate of 13.7% [16,17]. In the month of March 2020, northern regions accounted for 120% and 568% increases in mortality compared to the same period in 2019 [18].

The pandemic spread rapidly across Italy, and, by 8 March 2020, all regions had reported at least one COVID-19 case or death [19]. From the beginning of the pandemic, cities in the north and central Italy registered more than half of the total COVID-19 cases, with Lombardy (87,417 positive cases), Piedmont (30,314), Veneto (19,105), Tuscany (10,070, and Emilia-Romagna (27,611) recording the highest numbers of positive cases. In particular, the Lombardy region faced the most severe impacts on human lives, putting the region’s health system on the brink of a massive collapse [17].

For this study, major Italian metropolitan cities were explored to understand the interconnections between geographical location, number of COVID-19 cases, and social responses to the pandemic and locally enforced measures. A total of 10 cities were selected across multiple areas: (1) north: Milan, Venice, Turin, Bologna; (2) central: Florence, Rome; (3) south: Naples, Bari; and (4) islands: Palermo, Cagliari.

### 2.2. Data Collection

We collected a total of 4,227,882 COVID-related tweets between 2 March 2020 and 15 June 2020 with the use of the Twitter search application programming interface (api). For the search, we employed a set of Italian and English terms associated with COVID-19. Keywords included: COVID-19; Italy; Sentiment analysis; Twitter; Fear; Anger; Joy; Metropolitan cities; and Social network analysis. Twitter keywords consisted of a set of predefined coronavirus search key terms in both the English and Italian languages. Predefined Twitter search key terms/hashtags in English included: COVID-19, Coronavirus, CoronavirusOutbreak, coronavirusitaly, racism, COVID2019, COVID19italy, Flu, ItalyCoronavirus, Lombardy, Italyquarantine, quarantineItaly, and COVID. Predefined Twitter search key terms in Italian included: razzismo, Italiani all’estero, Influenza, Amuchina, Codogno, Contagiati, Contagio, Coronaviriusitalia, COVID19italia, COVID2019italia, Coronavirusitalia, CoronavirusItalla, Lombardia, zonarossa, focolai, and quarentena.

For data processing, we employed several Python libraries, including Tweepy, Pandas, and BeautifulSoup, that served as bases to create our own script. In particular, the Tweepy library was employed to provide API access to Twitter, while Panda served to handle the data frames, and BeautifulSoup to extract data from HTML and XML files. In addition, we used other libraries, such as re(regular expressions), json, sys, and datatime. Next, we extracted and sorted text and metadata from our Twitter search, paying special attention to possible misspellings.

### 2.3. Data Analysis

Twitter was our main platform for mining public sentiments. In fact, it provides access to a consistent number of public-generated posts and reactions regarding COVID-19. More specifically, our analysis considered sentiments related to fear, anger, and joy. The sentiment analysis was conducted using the NRC Emoticon Lexicon [20,21], a platform scoring 14,182 positive and negative unigrams. Based on this scoring system, we obtained a normalized score for each collected emotion. Finally, the mean sentiment score for fear, anger, and joy was computed across all tweets collected for a given date. The findings were then crossed analyzed with the total number of COVID-19-positive cases, national COVID-19 measures, and user characteristics. 

Figure 1 shows the research workflow metrics used to develop a sentiment analysis (SA). Over 4 million Twitter tweets related to fear, anger, and joy were analyzed for major Italian metropolitan cities during the first wave of the COVID-19 pandemic. The 10 cities included: (i) north: Milan, Venice, Turin, Bologna; (ii) central: Florence, Rome; (iii) south: Naples, Bari; and (iv) islands: Palermo, Cagliari.

To this end, we selected a number of socioeconomic indicators, such as (1) demographics, such as percentage of people aged 65+, total number of COVID-19 deaths, and cumulative number of COVID-19-positive cases, including percentage of people aged 65 or older, total number of COVID-related deaths, and cumulative number of COVID-19 cases; (2) environmental indicators, such as population density, industry manufacture expenditure, and industry services expenditure; and (3) economic indicators, including GDP per capita, unemployment rate, and intensive care unit (ICU) beds to explore the differences and capacities between the four Italian regions of Italy (north, central, south, and islands. A one-way ANOVA or *t*-test was performed to determine statistically significant differences between the mean values of different Italian regions and assess the influence of each selected indicator.

## 3. Results

Through analyses of sentiments for the COVID-19 Twitterverse at the global, national, regional, and city levels, analyses may reveal important intersections of public events, policies, and public reactions. Such focused analyses become important for the adaptation of health campaigns and policies to the specifics of a given locality. This study explored the impact of fear, anger, and joy during the first wave of COVID-19 in 10 Italian metropolitan cities. This analysis served to explore possible interconnections between such emotions and the enforcement of nation-wide measures to contain the pandemic. 

Scholars have found that mental health emerged during the COVID-19 outbreak, including stress, anxiety, depression, frustration, and uncertainty [22]. In relation to mass quarantines imposed in order to attenuate the spread of COVID-19, common psychological reactions of fear and pervasive community anxiety occurred, which are typically associated with disease outbreaks. These reactions increased with the escalation of new cases together with anxiety-provoking (mis)information provided by media. 

### 3.1. Sentiment Analysis 

Mean sentiment analysis at the national level is presented below. Exact information for the 10 selected cities is presented in Appendix A. Overall, for fear and anger, our results show that the initial response (March–April) was volatile and trending at higher values, with its highest level registered on 25 April 2020. After this date, values decreased, reaching a new low on 4 May 2020, which corresponded with the beginning of Phase II of the pandemic. Later, we registered an increased value for both the fear and anger sentiments. Contrarily, the sentiment values for joy were higher at the early stages of the pandemic and dropped as time passed. Overall, the response for joy was volatile, with a mean value of approximately 1.3%.

#### 3.1.1. Fear

Figure 2 shows the mean emotion for fear varying over time. Out of the total collected tweets, the highest mean was found to be in the month of April at 2.39% when compared to May at 2.35%, and March at 2.30%, respectively. In this corpus of Italian tweets about COVID-19, 2.34% were related to fear.

#### 3.1.2. Anger

Figure 3 shows the mean emotion for anger varying over time. Out of the total collected tweets, our analysis shows that the highest frequency of fear sentiments (1.29%) was registered during the month of April 2020, with an average value of 1.19%. A slightly lower occurrence was registered in the months of March (1.13%) and May (1.23%).

#### 3.1.3. Joy

The highest incidence of joy tweets was registered in the month of March 2020, with a frequency of 1.48% of the total collected tweets, compared to April (1.45%) and May (1.26%). Figure 4 shows a total occurrence of 1.44% tweet frequency associated with joy. 

### 3.2. Cross-City Variations

A two-tailed, two-sampled *t*-test was conducted for each city and emotion to assess possible differences between the mean levels of sentiments in each city under study. From our results, it appeared that sentiments were similarly distributed across cities. With an alpha level value of 0.5R. A cross-city comparison for the three sentiments, fear, anger, and joy, is presented below.

#### 3.2.1. Fear

Table 1 shows the *p*-values of the cross-city comparison for tweets related to fear. When comparing the results between the mean levels of fear, all but three city pairs’ *p*-values were statistically significant (Table 2). The city pair values that proved to be insignificant because their values were within the range of −1.96 and 1.96, were Bologna–Cagliari (−1.802), Florence–Rome (−1.898), and Naples–Palermo (−1.703). Hence, this provides evidence that geographical location influenced most fear sentiment levels. 

#### 3.2.2. Anger 

Table 3 shows the *p*-values of the cross-city comparison for anger tweets. The results for the cross-city comparison for anger show that all but five city pairs recorded statistically significant values (>0.5) (Table 4). 

The difference between the sentiment mean variable for anger in the cities of Bari and Bologna was 10.29 times smaller than expected, while the anger t-statistic values demonstrated a difference in the means which was approximately larger or smaller than expected based on chance. Insignificant values due to the t-statistic, meaning between −1.96 and 1.96, were found between Bari–Bologna (1.695), Bari–Turin (−1.841), Milan–Naples (−0.532), Milan–Venice (1.170), and Naples–Venice (1.326). This provides evidence that geographical location influenced most anger sentiment levels.

#### 3.2.3. Joy

Table 5 shows the *p*-values of the cross-city comparison for joy tweets. When comparing the results between the mean levels of joy, all but two city pairs’ *p*-values were statistically significant (Table 6). The city pair values that proved to be insignificant because their values were within the range of −1.96 and 1.96 were Bari–Rome (0.368) and Florence–Naples (−0.478). This provides evidence that geographical location influenced most joy sentiment levels.

## 4. Discussion

Previous studies conducting a sentiment analysis of the tweets collected during epidemics have shown that fear generally increases during the pre-pandemic period [23,24], becoming a dominant sentiment as users seek who/what to blame for the threats posed to their lives, and eventually decreases over the course of the epidemic, as cases and deaths decrease and more containment measures are put in place [25,26]. In contrast, anger increases as people begin to organize appraisals of who and what to blame for the threats posed to their lives. The tweets in this corpus did not reveal this pattern, at least not within the time span covered in this study. Apart from Sicily COVID-19 tweets, our data showed different patterns for the first wave of the COVID-19 pandemic in Italy. Tweets collected in north and central regions demonstrated rising trends for fear throughout the entire first wave of the pandemic, even when COVID-19 cases started to drop. This trend is possibly associated with uncertainties surrounding the post-pandemic social and economic recovery, including a “new normal” and the prospect of further COVID-19 waves. While fear is a primary motivator, studies showed that it only plays functional roles in health when coupled with trust in the existing healthcare infrastructure. In our study, fear presented the highest values during the last week of April in Italy’s central regions, when several businesses were retrieving their operations. Like previous studies [27], we found that fear was central to social media communication. COVID-19 tweets, in fact, have rapidly spread within topic networks, with fear and stress being main topic themes, especially in association with China [28]. According to another study, over half of COVID-19-related tweets posted in the month of January, 2020, contained fear emotions. This frequency dropped in early April to under 30%, while anger went from 13–14% in January, 2020, to over 30% in early April [23]. 

Notably, our results show an increase in anger tweets from the last week of May in the islands, following the lift of lockdown measures and consequent re-opening of traveling. A study on anger found that this sentiment is far more important in mobilizing public and political action, as well as triggering the propagation of negative news among society, compared to the other sentiments [29]. Fear plays similar influential roles in social media, being uniquely capable of creating cascading and contagious communications [30]. For example, another study on sentiment analysis conducted in China showed similar shifts in the emotions over the course of the pandemic [31]: “as the COVID-19 epidemic began to spread throughout the country after 20 January 2020, the public eased their concerns and fears caused by their uncertainty toward and ignorance of the epidemic and responded to the epidemic with a more objective attitude” [24].

Findings demonstrated a connection between: Exposure to news or events and how this produces effects on the feelings of the population. Scholars have shown that when everyday news was perceived as more negative, subjects experienced more negative effects and fewer positive effects. Additionally, research indicates that people report more negative effects when negative news items are personally relevant [32]. Other studies have underlined the role of emotion in susceptibility to believing fake news. There is a connection between emotion and fake news in which self-reported use of emotion is positively associated with belief in fake news, inducing reliance on emotion and a greater belief in fake news stories compared to a control group or to inducing reliance on reason [33]. 

Psychological processes (emotional, perceptive, and cognitive) produce effects on the manifestation of feelings (positive and negative) when exposed to news events. Studies have shown that emotion has a substantial influence on the cognitive process in humans, including perception, attention, learning, memory, reasoning, and problem solving. As a result, emotion has a strong connection to cognitive influences, perception, and memory, which possibly played a role during the pandemic [34]. 

The study explored 10 metropolitan case study cities based on geography in the north, center, south, and island regions of Italy. We selected 10 Italian cities as a representative sample of a population, seeking to accurately reflect the characteristics of the larger group. Hence, the 10 representative selected cities that were strongly hit during the first wave of the pandemic are considered to have strong political and geographic territorial influences that were reflected in their online expressions of emotion [17,19]. Societal and cultural factors associated with the Italian context, such as traditions, beliefs, perceptions, attitudes, and behaviors, may correlated with the type of sentiment and the way emotion is expressed by the population based on time. Although sentiment analysis is considered an effective method for identifying people’s opinions in mass, the classification errors of standard systems affect the results of the sentiment analysis and may reflect inaccurate analysis. Also, sentiment analysis provides some sense of people’s opinions, and the biases reflected in tweets may in turn mislead users and policy-makers and cause them to make erroneous decisions. Furthermore, societal cultural factors in Italy may have had an impact on their emotions based on socio-economic factors such as age, education, literacy, political preference, viewpoint, family member’s health, own health, or economic/financial burden during the COVID-19 pandemic [35,36].

## 5. Conclusions

To the best of our knowledge, this is the first study analyzing spatial differences in the social response to COVID-19 in Italy using Twitter data. The study offers insights into the distribution of emotions (fear, anger, and joy) related to the spread of COVID-19 across cities located in the north, center, south, and islands of Italy. The study employed a sentiment analysis to further the understanding of emotions contained in user tweets in response to specific measures and milestones of the lockdown and subsequent phases during the months of March to June 2020. Findings demonstrate a connection between exposure to news and/or significant policy measure events, and how this produces effects on the feelings of the population. In addition, the emotional, perceptive, and cognitive psychological processes produce effects on the manifestation of feelings (positive and negative) when exposed to news events. 

Scholars have found that there was a rapid worldwide spread of deleterious socioeconomic and psychological impact of the COVID-19 pandemic. A number of psychological problems and important consequences in terms of mental health, such as stress, anxiety, depression, frustration, and uncertainty, during COVID-19 infection, have been additionally documented together with the most relevant psychological reactions of the general population related to the COVID-19 pandemic. The results of this study can allow decision makers to take a step back and re-design institutional communication strategies related to changes in health policies that are aimed at generating positive feelings in the population. Understanding the effects that information produces on the perception and feelings of the population regarding certain events that affect them provides a potential information resource for adjusting health campaigns. Finally, strategies can be proposed that mitigate the appearance of negative feelings in the population as a result of being exposed to certain events/news associated with new political initiatives. 

Limitations of this study pertain to the selection of the sample size, including the platform used, and the data collection timeline. Moreover, t-statistics represent a relatively static representation of regional differences. Multivariate time-series statistics might provide better a more dynamic approach to understanding such processes. 

Future studies should analyze the specific numerical levels of the sentiment for emotions, alongside any theorized or developed baselines for certain levels of sentiment. Such an analysis could find commonalities in emotional word usage related to statistically significant differences in sentiment levels. Moreover, future research should look more at why some people are more affected by negative news than other news. Finally, in order to develop better predictive models in the future, we suggest including variables that further reflect lockdown levels of individual regions over time. Such variables could be made from close examination of adopted policies, as well as be derived from citizen–government perception–behavior surveys, environmental impacts, finance, and/or management. 

## Figures and Tables

**Figure 1 ijerph-19-07720-f001:**
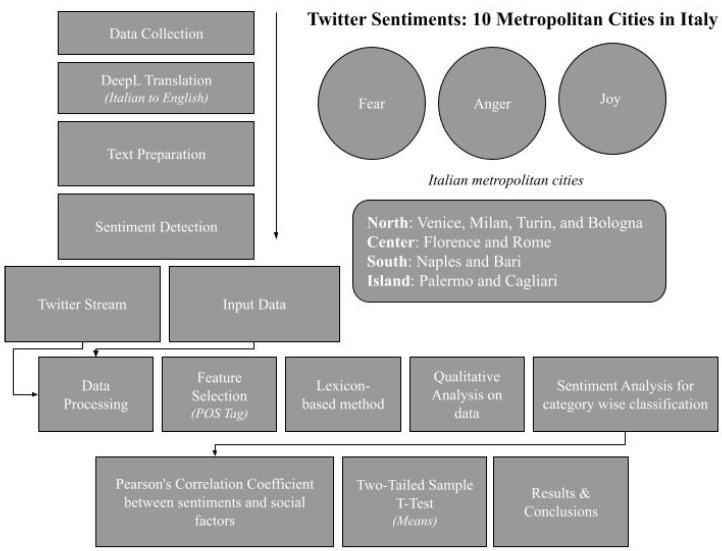
Research work flow model.

**Figure 2 ijerph-19-07720-f002:**
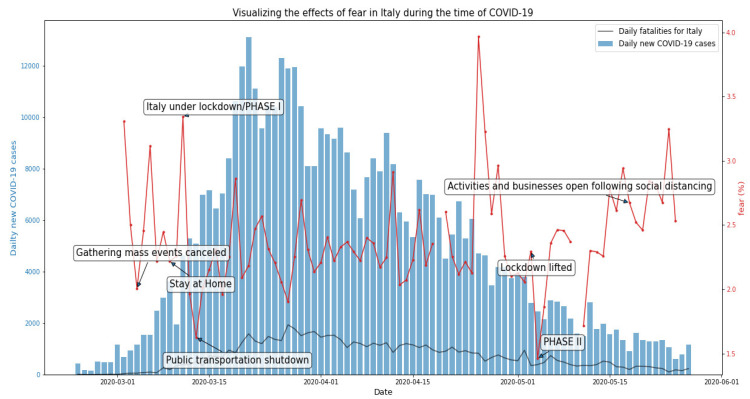
Fear-related tweets in Italy during the first wave of the COVID-19 pandemic.

**Figure 3 ijerph-19-07720-f003:**
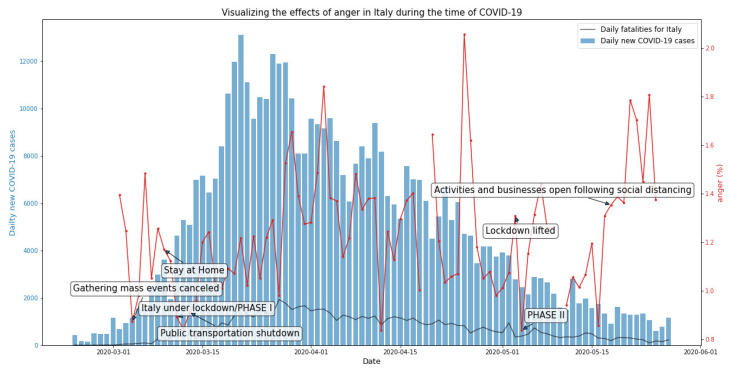
Anger-related tweets in Italy during the first wave of the COVID-19 pandemic.

**Figure 4 ijerph-19-07720-f004:**
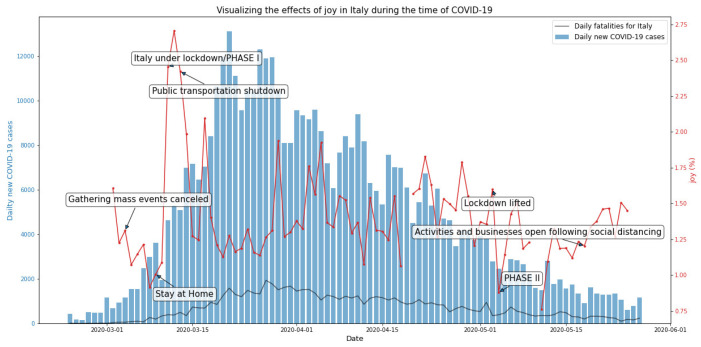
Joy-related tweets in Italy during the first wave of the COVID-19 pandemic.

**Table 1 ijerph-19-07720-t001:** Fear *p*-values.

Fear T-Stats	Bari	Bologna	Cagliari	Florence	Milan	Naples	Palermo	Rome	Turin
Bologna	9.56 × 10^−3^								
Cagliari	2.21 × 10^−4^	7.15 × 10^−2^ *							
Florence	7.68 × 10^−54^	2.91 × 10^−126^	8.27 × 10^−79^						
Milan	7.93 × 10^−7^	1.55 × 10^−30^	4.28 × 10^−21^	1.64 × 10^−121^					
Naples	3.52 × 10^−19^	1.36 × 10^−58^	3.23 × 10^−38^	5.30 × 10^−35^	2.02 × 10^−29^				
Palermo	1.80 × 10^−10^	4.54 × 10^−27^	8.37 × 10^−24^	3.24 × 10^−23^	1.00 × 10^−4^	8.86 × 10^−2^ *			
Rome	8.10 × 10^−59^	5.88 × 10^−168^	1.05 × 10^−85^	5.76 × 10^−2^ *	0	2.00 × 10^−65^	2.23 × 10^−25^		
Turin	2.44 × 10^−4^	4.72 × 10^−19^	7.16 × 10^−16^	5.22 × 10^−98^	1.18 × 10^−2^	2.96 × 10^−25^	1.96 × 10^−6^	2.46 × 10^−200^	
Venice	4.22 × 10^−28^	1.78 × 10^−73^	6.98 × 10^−49^	6.99 × 10^−13^	4.84 × 10^−45^	1.38 × 10^−5^	6.34 × 10^−6^	1.14 × 10^−15^	5.31 × 10^−40^

Note: characteristics: total COVID-19 cases: total number of COVID-19 cases; daily COVID-19 cases: total number of COVID-19 cases per day; total COVID-19 deaths: total number of deaths; daily COVID-19 deaths: total number of deaths per day; anger: total number of Twitter tweets related to anger; joy: total number of Twitter tweets related to joy; fear: total number of Twitter tweets related to fear. * means that *p*-value is less than 0.05 but more than or equal to 0.01.

**Table 2 ijerph-19-07720-t002:** Fear t-static values.

Fear T-Stats	Bari	Bologna	Cagliari	Florence	Milan	Naples	Palermo	Rome	Turin
Bologna	−2.591	-	-	-	-	-	-	-	-
Cagliari	−3.694	−1.802 *	-	-	-	-	-	-	-
Florence	15.463	23.926	18.831	-	-	-	-	-	-
Milan	4.938	11.491	9.432	−23.463	-	-	-	-	-
Naples	8.954	16.148	12.939	−12.345	11.263	-	-	-	-
Palermo	6.378	10.777	10.063	−9.927	3.891	−1.703 *	-	-	-
Rome	16.195	27.691	19.676	−1.898 *	57.531	17.086	10.415	-	-
Turin	3.668	8.920	8.071	−21.020	−2.519	−10.384	4.758	−30.224	-
Venice	10.996	18.142	14.712	−7.180	14.088	4.347	4.515	−8.012	13.240

Note: characteristics: total COVID-19 cases: total number of COVID-19 cases; daily COVID-19 cases: total number of COVID-19 cases per day; total COVID-19 deaths: total number of deaths; daily COVID-19 deaths: total number of deaths per day; anger: total number of Twitter tweets related to anger; joy: total number of Twitter tweets related to joy; fear: total number of Twitter tweets related to fear. * means that *p*-value is less than 0.05 but more than or equal to 0.01.

**Table 3 ijerph-19-07720-t003:** Anger *p*-values.

Fear T-Stats	Bari	Bologna	Cagliari	Florence	Milan	Naples	Palermo	Rome	Turin
Bologna	7.830 × 10^−25^	-	-	-	-	-	-	-	-
Cagliari	5.292 × 10^−29^	6.132 × 10^−4^	-	-	-	-	-	-	-
Florence	6.632 × 10^−11^	2.595 × 10^−110^	4.667 × 10^−79^	-	-	-	-	-	-
Milan	4.309 × 10^−2^	2.601 × 10^−87^	1.024 × 10^−57^	6.280 × 10^−22^	-	-	-	-	-
Naples	9.007 × 10^−2^ *	4.146 × 10^−69^	4.051 × 10^−52^	3.790 × 10^−17^	5.945 × 10^−1^ *	-	-	-	-
Palermo	2.037 × 10^−14^	8.574 × 10^−96^	7.888 × 10^−79^	1.196 × 10^−2^	2.947 × 10^−21^	4.615 × 10^−19^	-	-	-
Rome	1.307 × 10^−9^	1.267 × 10^−142^	1.333 × 10^−83^	2.025 × 10^−2^	4.162 × 10^−84^	5.727 × 10^−25^	5.877 × 10^−6^	-	-
Turin	6.569 × 10^−2^ *	7.627 × 10^−38^	4.643 × 10^−34^	1.977 × 10^−44^	4.526 × 10^−20^	2.052 × 10^−11^	4.790 × 10^−39^	5.676 × 10^−79^	-
Venice	1.717 × 10^−2^	1.302 × 10^−64^	1.893 × 10^−52^	8.071 × 10^−10^	2.419 × 10^−1^ *	1.849 × 10^−1^ *	2.760 × 10^−13^	1.336 × 10^−9^	5.429 × 10^−12^

Note: characteristics: total COVID-19 cases: total number of COVID-19 cases; daily COVID-19 cases: total number of COVID-19 cases per day; total COVID-19 deaths: total number of deaths; daily COVID-19 deaths: total number of deaths per day; anger: total number of Twitter tweets related to anger; joy: total number of Twitter tweets related to joy; fear: total number of Twitter tweets related to fear. * means that *p*-value is less than 0.05 but more than or equal to 0.01.

**Table 4 ijerph-19-07720-t004:** Anger *t*-statistic values.

Fear T-Stats	Bari	Bologna	Cagliari	Florence	Milan	Naples	Palermo	Rome	Turin
Bologna	−10.293	-	-	-	-	-	-	-	-
Cagliari	−11.182	−3.426	-	-	-	-	-	-	-
Florence	6.530	22.335	18.862	-	-	-	-	-	-
Milan	2.023	19.831	16.044	−9.626	-	-	-	-	-
Naples	1.695 *	17.582	15.213	−8.420	−0.532 *	-	-	-	-
Palermo	7.650	20.785	18.825	2.513	9.468	8.923	-	-	-
Rome	6.068	25.480	19.426	−2.322	19.432	10.321	−4.531	-	-
Turin	−1.841 *	12.864	12.178	−13.986	−9.175	−6.702	−13.077	−18.821	-
Venice	2.383	16.981	15.260	−6.144	1.170 *	1.326 *	−7.306	−6.063	6.894

Note: characteristics: total COVID-19 cases: total number of COVID-19 cases; daily COVID-19 cases: total number of COVID-19 cases per day; total COVID-19 deaths: total number of deaths; daily COVID-19 deaths: total number of deaths per day; anger: total number of Twitter tweets related to anger; joy: total number of Twitter tweets related to joy; fear: total number of Twitter tweets related to fear. * means that *p*-value is less than 0.05 but more than or equal to 0.01.

**Table 5 ijerph-19-07720-t005:** Joy *p*-values.

Fear T-Stats	Bari	Bologna	Cagliari	Florence	Milan	Naples	Palermo	Rome	Turin
Bologna	6.10 × 10^−94^	-	-	-	-	-	-	-	-
Cagliari	3.29 × 10^−27^	1.26 × 10^−9^	-	-	-	-	-	-	-
Florence	4.69 × 10^−3^	4.16 × 10^−116^	3.80 × 10^−24^	-	-	-	-	-	-
Milan	3.29 × 10^−45^	6.96 × 10^−48^	9.81 × 10^−3^	1.92 × 10^−73^	-	-	-	-	-
Naples	8.87 × 10^−4^	7.99 × 10^−131^	7.99 × 10^−25^	0.632 *	5.05 × 10^−116^	-	-	-	-
Palermo	9.80 × 10^−6^	2.34 × 10^−70^	1.26 × 10^−14^	1.44 × 10^−2^	2.95 × 10^−23^	2.25 × 10^−2^	-	-	-
Rome	0.713 *	5.71 × 10^−215^	2.27 × 10^−43^	1.29 × 10^−9^	0	1.78 × 10^−18^	6.21 × 10^−13^	-	-
Turin	0.044	6.88 × 10^−148^	7.48 × 10^−30^	0.133	4.73 × 10^−153^	0.018	1.40 × 10^−4^	1.08 × 10^−8^	-
Venice	4.42 × 10^−21^	8.76 × 10^−51^	5.19 × 10^−6^	5.51 × 10^−20^	1.13 × 10^−6^	8.48 × 10^−23^	3.54 × 10^−7^	1.50 × 10^−70^	5.92 × 10^−32^

Note: characteristics: total COVID-19 cases: total number of COVID-19 cases; daily COVID-19 cases: total number of COVID-19 cases per day; total COVID-19 deaths: total number of deaths; daily COVID-19 deaths: total number of deaths per day; anger: total number of Twitter tweets related to anger; joy: total number of Twitter tweets related to joy; fear: total number of Twitter tweets related to fear. * means that *p*-value is less than 0.05 but more than or equal to 0.01.

**Table 6 ijerph-19-07720-t006:** Joy t-statistics values.

Fear T-Stats	Bari	Bologna	Cagliari	Florence	Milan	Naples	Palermo	Rome	Turin
Bologna	−20.584	-	-	-	-	-	-	-	-
Cagliari	−10.809	6.074	-	-	-	-	-	-	-
Florence	−2.828	22.927	10.142	-	-	-	-	-	-
Milan	−14.126	14.548	2.583	−18.138	-	-	-	-	-
Naples	−3.324	24.369	10.295	−0.478 *	22.905	-	-	-	-
Palermo	−4.422	17.743	7.712	−2.448	9.938	−2.282	-	-	-
Rome	0.368 *	31.393	13.828	6.069	63.779	8.771	7.197	-	-
Turin	−2.015	25.938	11.358	1.504	26.367	2.365	3.808	−5.718	-
Venice	−9.425	14.994	4.558	−9.154	4.868	−9.830	−5.092	−17.768	−11.767

Note: characteristics: total COVID-19 cases: total number of COVID-19 cases; daily COVID-19 cases: total number of COVID-19 cases per day; total COVID-19 deaths: total number of deaths; daily COVID-19 deaths: total number of deaths per day; anger: total number of Twitter tweets related to anger; joy: total number of Twitter tweets related to joy; fear: total number of Twitter tweets related to fear. * means that *p*-value is less than 0.05 but more than or equal to 0.01.

## Data Availability

Data supporting reported results can be found, including links to publicly archived datasets analyzed or generated during the study. All code used in this study to generate the Twitter translation and sentiment analysis is freely available in our GitHub repository: https://github.com/HDMA-SDSU/Translate-Tweets (accessed on 16 June 2022). Please visit our Social Response to COVID-19 in Italy Story Maps here: https://storymaps.arcgis.com/stories/74c499d5ac0a46ffbbc2b28acfa05102 (accessed on 16 June 2022).

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
