# Peer review of "Social Network Analysis of COVID-19 Sentiments: 10 Metropolitan Cities in Italy"

_ijerph, 2022, doi:10.3390/ijerph19137720_

Round 1
Reviewer 1 Report
Dear editor and esteemed authors,
Thank you for giving me the opportunity to review the manuscript submitted to IJERPH. The article represents an interesting contribution to the literature, applying social network analysis to examine the network of feelings in a virtual communication platform such as Twitter.
The article is relevant because, to a certain extent, it shows that the feelings of the population vary (as expected) in the face of an exceptional global emergency situation such as a COVID-19 pandemic.
Also because it explores how changes in certain types of events related in this case to changes in public policies that affect the population, can directly affect the feelings that the population expresses through digital communication networks.
For these reasons, the results of this research may have implications in several lines that are described below:
- Re-design political communication strategies related to changes in health policies that are aimed at generating positive feelings in the population.
- Understand the effects that information produces on the perception and feelings of the population regarding certain events that affect them.
- Propose strategies that mitigate the appearance of negative feelings in the population as a result of being exposed to certain events associated with new political initiatives.
Although I consider that the work is of interest and methodologically robust, there are some aspects that I consider should be improved:
- The authors must better justify the relationship between exposure to news or events, and how this produces effects on the feelings of the population.
- Explain more clearly what are the psychological processes (emotional, perceptive and cognitive) that produce effects in the manifestation of feelings (positive and negative) when exposed to news events. Studies on this topic should be reviewed and cited in the introduction and in the discussion.
- Justify in greater detail the choice of the evaluated contexts (cities and some island, and the country).
- Explain whether there are societal cultural factors (associated with the Italian context) that may be affecting the type of feeling and the way it is expressed by the population. That is, if the cultural characteristics of the context (individualism versus collectivism) can help explain the results of this research.
Introducing these changes is important to help interpret the results and conclusions of the research.
Best regards,
Author Response
Thank you again for your wonderful feedback. Everything you provided was very useful for the paper. I really appreciate it. I have integrated your comments into the paper.
- Done.
- Done.
- Impactful and politically powerful metropolitan cities of Italy. Representation of the overall population.
- Great feedback. Very helpful. I have added further information to the introduction and discussion.
Reviewer 2 Report
The topic of this study is very interesting. The manuscript is well written. However, in order to improve the quality of the manuscript, my suggestions are given below.
1) In the introduction section please mention the benefits of your research, highlight your contribution using bullets or points and also add the motivation of this research study. 2) The introduction section is very brief, you should add a diagram of "sentiment analysis, and explain its basics using the diagram.
3) Please include the organization of the paper at the end of the introduction section. For example; Overall, this study is organized as follows. As a reference study, you can check the format for the writing of the paper using the below ref:
F. and G. S. Choi, "Hotspots Analysis Using Cyber-Physical-Social System for a Smart City," in IEEE Access, vol. 8, pp. 122197-122209, 2020.
4) The Related work section is missing. Please add the most recent studies to this section. In addition, please add a comparison table and compare the most recent studies along with advantages and disadvantages. For the comparison table, you can check the below study.
5)In the conclusion section please mention the future research of your study.
Author Response
Thank you for your great feedback. I really appreciate it.
- Done. I did not include the bullets to save space. I added the information in the text.
2. Done see Figure 1
- Done. I added a few studies without the need to add a table due to the limited amount of space.
- Done.
Reviewer 3 Report
Statistics of the raw data grabbed from Twitter need to be more detailly presented. Since dealing with text mining techniques, details of method need to be elaborated, not just mentioning tools involved in revealing sentiment. Every tools involved in the work has their own trade off including NRC Emotion lexicon, it is better to deliver sort of validation metrics showing the confidence of the presented result.
Author Response
Thank you for your feedback. I really appreciate it. I have made the necessary changes.
Reviewer 4 Report
This paper deals with an extensive analysis of emotions contained in Twitter data during the first wave of the pandemic. The authors present sentiment analysis results in 10 Italian cites. Overall, the issue towards this study is of practical significance. However, the following comments should be addressed while preparing the revision.
1. In Abstract, the background and motivation of this study are not clearly stated.
2. The contributions of this study need to be refined to highlight the significance of this work.
3. In Section 3.2, some technical details regarding to data analysis need to be supplied.
4. The reason why the authors choose p-values and T-static as metric need to be explained.
5. Network analysis is widely studied in different real applications, and the following studies regarding it are highly related to this paper and they may be useful for this paper, e.g., “Improving network topology-based protein interactome mapping via collaborative filtering”, “Efficiently detecting protein complexes from protein interaction networks via alternating direction method of multipliers”, “A novel approach to large-scale dynamically weighted directed network representation”, and “NeuLFT: a novel approach to nonlinear canonical polyadic decomposition on high-dimensional incomplete tensors”.
Author Response
- The abstract has been edited based on recommendations.
- Done.
- Done.
- Done.
- Thank you for useful articles.
Round 2
Reviewer 2 Report
The authors has incorporated all necessary changes.